# Integrating the Sufficiency Economy Royal Philosophy and Participatory Action Research Approach to Promote Self-Care for Stroke Prevention in Selected Communities of Southern Thailand

**DOI:** 10.3390/healthcare12141367

**Published:** 2024-07-09

**Authors:** Chutarat Sathirapanya, Jamaree Trijun, Pornchai Sathirapanya

**Affiliations:** 1Department of Family Medicine and Preventive Medicine, Faculty of Medicine, Prince of Songkla University, Hat Yai, Songkhla 90110, Thailand; 2Department of Pharmacy, Khaochaison Hospital, Phatthalung Provincial Public Health Office, Khaochaison, Phatthalung 93130, Thailand; jampharmacy@hotmail.com; 3Department of Medicine, Faculty of Medicine, Prince of Songkla University, Hat Yai, Songkhla 90110, Thailand; sporncha@medicine.psu.ac.th

**Keywords:** stroke prevention, economy, participatory action, self-care, community health

## Abstract

(1) Introduction: Effective control of stroke risk factors can reduce stroke incidence. Motivation for participatory action of community dwellers to practice self-care to modify stroke risk after providing them with knowledge of stroke risk factors is considered useful under a situation of limited healthcare resources. This study aimed to evaluate the outcomes of integrating the sufficiency economy philosophy (SEP), a royal economic philosophy in Thailand, and the participatory action research (PAR) approach on stroke risk factors control among selected communities. (2) Methods: Villagers who had medium to high stroke risk from two provinces with leading stroke incidences in southern Thailand were invited to participate in an eight-month SEP-PAR program conducted in 2019. Group meetings among the study participants, local healthcare providers, the researchers, and relevant stakeholders in the communities were held to co-design a health behaviors program targeting lower waist circumference (WC), body weight (BW), blood pressure (BP), fasting blood sugar, blood lipids, and smoking and alcohol consumption rates. Follow-up physical measurements and blood tests were compared with the baseline results for significant differences by descriptive statistics (*p* < 0.05) using the R program. (3) Results: Of 126 participants, 75.4% were female. Moderate and high stroke risk levels were found in 58.2% and 19.8%, respectively. Elevated baseline WC, BW, BP, and blood test results were found in 50–80% of the participants. The co-designed health behaviors in this study were dietary control, regular exercise, relieving psychological stress, and stopping smoking and alcohol consumption. Overall, the participants had significant adherence to the co-designed health behaviors. At the end of the program, the follow-up tests showed significant reductions in BW, BP, fasting blood sugar, and lipids, but not in WC. (4) Conclusions: A combined SEP and PAR approach was effective for stroke risk factors control among the community dwellers. Motivation for self-care is a significant strategic outcome expected of this approach. Longer follow-up studies in larger populations are needed.

## 1. Introduction

### 1.1. Background

Stroke is a major global health concern, as the second leading cause of death and the third leading cause of death and disability combined assessed by disability-adjusted life years (DALYs) lost. The estimated global cost of stroke treatment is over USD 721 billion (0.66% of the global GDP) [1]. Hypertension (HT), diabetes mellitus (DM), dyslipidemia (DLP), central or abdominal obesity measured by waist circumference (WC), and smoking are known major risk factors for stroke, coronary heart disease, and peripheral vascular diseases [2,3]. Male gender, high-salt diet, and family history of stroke were major stroke risk factors, while regular exercise was a protective factor found in a study [4]. A population fraction attributable study of stroke risk factors conducted in Thailand based on two national non-communicable disease surveillance systems found that approximately 41.6% of strokes in the Thai population were related to HT, while smoking carried the highest population fraction attributable to stroke incidence in those aged 15–34 years [5]. This study advocated for intense HT control among the general population, while emphasizing smoking cessation among the younger population to reduce national stroke incidence [5]. Currently, both pharmacological treatments and neuroradiological interventions have been intensively investigated for their benefits in reducing stroke morbidity and mortality [6,7,8]. However, stroke risk factors significantly affect stroke incidence. Except for age, other stroke risk factors are modifiable by practicing healthy behaviors [9,10,11]. A recent study analyzed current and predicted future trends of attributable stroke risk factors and concluded that two behavioral risks, i.e., smoking and high dietary sodium, and five cardio-metabolic factors, i.e., elevated systolic blood pressure, low-density lipoprotein cholesterol, fasting plasma glucose, body mass index (BMI), and impaired renal function, are currently and will be the major contributors to increased ischemic stroke burden in the future [12]. Previous studies also found that under-controlled cardiovascular risk factors predispose patients to high stroke morbidity and mortality [13,14]. In Thailand, although overall stroke mortality has been decreasing (on average 10.7% per year) due to advances in acute stroke treatment, the reported annual incidence of all strokes among people aged ≥15 years increased from 2008.9 per 100,000 in 2016 to 2245.1 per 100,000 in 2018 [15]. Since aging is also a risk factor of stroke, an increasing number of older people in a country will affect the national stroke incidence. Moreover, aging is one of the various factors determining functional recovery and quality of life (QoL) among stroke survivors [16]. According to a recent report from the Department of Older Persons, Ministry of Social Development and Human Security, Thailand, the number of Thai people aged >60 years was 12.8 million (19.4% of the total population) at the end of June, 2023 [17]. Hence, Thailand and other countries with similar trends of aging populations can be expected to experience higher stroke incidences, causing lower productivity and QoL of the people in the future. Meanwhile, due to global economic constraints, the available healthcare budget and related resources are very limited in Thailand. To counter this problem, self-care practices supported by providing sufficient health knowledge and advice to improve individual health behaviors should be encouraged. Hence, under the current economic constraints of the nation, self-reliant learning to gain health knowledge, forming positive attitudes from the available health information sources, and subsequent modification of personal health behaviors are methods of promoting self-care practice, hopefully helping to reduce the disparity between the available and required stroke service resources. A recent study reported that the level of self-efficacy (OR: 0.753; *p* = 0.048) and self-care practices (OR: 0.817; *p* = 0.018) were inversely related to stroke incidence, while stroke risk scores (OR: 3.513; *p* = 0.005) directly predicted it. The same study also found a positive association between the influential outcomes of both self-efficacy and self-care in reducing stroke incidence [18]. Encouraging self-efficacy and self-care among community members for reducing stroke incidence by controlling active risk factors through self-directed methods is a potentially cost-effective healthcare approach under the current economic situation. Although an appropriate healthcare budget allocation to eliminate the healthcare disparity between urban and rural communities, as in our study context, has been encouraged [19], the process needs more time before the expected outcomes can come to fruition. We also believe that the active participation of people with their self-designed behaviors program, based on a clear understanding of the achievable health benefits, could bring about sustainable adherence to the program.

### 1.2. Theoretical Framework and Concept

In 1974, His Majesty the late King Bhumibol Adulyadej of Thailand initiated and introduced the sufficiency economy philosophy (SEP) for Thai people to live economically in the conditions of economic constraint and inequality. Hence, SEP is originally a socioeconomic concept. It principally focuses on maintaining a balance between household incomes and expenses to ensure household economic security. The key principle of SEP suggests that the Thai people should maintain the economic necessities of living without seeking to possess inessential things to keep a balanced household financial status [20]. His Majesty believed that the rapid and disproportionate economic growth of urban over rural communities could cause economic inequality to which the poor were more vulnerable.

The core concept of SEP consists of three interrelated major elements and two underlying conditions. The three major elements are (1) moderation in living or “moderation”, (2) generation of reasonable thoughts or “reasonableness”, and (3) forming careful behaviors for self-protection from economic threats or “prudence”. These three elements require two underlying conditions: (1) knowledge of an economical lifestyle and (2) Buddhist moral teachings such as kindness and compassion [21,22]. By having a good understanding or knowledge of the essentials for an individual’s daily living integrated with Buddhist moral teachings, SEP encourages people to reasonably plan to live moderately and develop self-protection behaviors against economic threats. SEP also encourages those who have achieved their financial security goals to provide support to underrepresented people for achieving better financial standards and establishing socio-economic equity in society as well. Basically, SEP encourages self-reliant household financial balance through moderate and reasonable utilization of available household resources first and then compassionate support of the underprivileged. Community practices based on SEP could reduce the economic gap in a society and have a positive impact on the economic sustainability of a country [23].

Due to the economic basis of SEP, the application of SEP concepts to healthcare management has been limited. However, in the situation of limited healthcare resources caused by national financial constraints needing mindful health budget allocations, the concepts of SEP, including the three major elements supported by the two basic conditions mentioned, have been considered applicable in some situations [24,25,26]. The conceptual framework of SEP application to community healthcare in terms of stroke risk reduction in this study is shown below (Figure 1). Additionally, facilitating active participatory action among community members to practice self-care could be another method to enhance effective healthcare with lower healthcare resource consumption [27,28,29,30]. It has been suggested that facilitation of healthy behaviors in a community could be conducted by the community dwellers themselves after realization of the risks of developing diseases [24,25,26,27]. Therefore, we suggested that the concepts of SEP and participatory action could be integrated because they shared the initial step of knowledge provision to the community; thus, reasonable thought formation, and active participatory actions would enhance the practice of self-care mindfully and moderately to create self-protecting behaviors or lifestyles, which follow the major elements of SEP [31,32]. Furthermore, those who achieved the health goal of controlling stroke risk factors by self-care could voluntarily encourage and facilitate non-achievers to practice self-care for modifying health behaviors compassionately based on the moral teachings (Figure 2). The SEP concept has been included in Thai national development schemes and associated national healthcare programs since 2002 [28]. Moreover, according to all recent Thai National Economic and Social Development Plans, the application of SEP is suggested as a strategic method to encourage self-care practices among Thai people. In this regard, the provision of sufficient health knowledge and close supervision by healthcare providers (HCPs) are fundamentally significant for integrating a combined SEP and participatory action research (PAR) approach [28].

### 1.3. Rationale and Objectives of the Study

In this study, we aimed to investigate the outcomes of SEP combined with the PAR method in controlling stroke risk factors. We thought that the PAR method would facilitate sustainable adherence to the co-designed health behavior program because of the participants’ perceptions of partnership in planning the program [29,30]. The health behaviors included in the program were delineated according to a consensus generated during group meetings which included the study participants, local healthcare workers, and representatives of various socio-cultural sectors. We evaluated the beneficial outcomes of the combined SEP and PAR method in controlling stroke risk factors through follow-up assessments of waist circumference (WC), body mass index (BMI), blood pressure (BP), fasting blood sugar (FBS), and blood lipid levels at the end of the program.

To date, few studies have applied the concept of SEP to healthcare, particularly in stroke risk factors control. We decided to study the outcomes of applying SEP and the PAR approach in stroke prevention because a stroke usually leads to long-term physical disability and high psychological impacts, which have a greater influence on the QoL of stroke survivors than other cardiovascular diseases. We also believed that the principles of SEP combined with the PAR approach in stroke prevention could encourage the community members to develop self-efficacy in performing self-care consistently, thus leading to lower governmental budgetary requirements for providing stroke prevention services. Practicing self-care, especially health behavior modification, is a principal measure of primary stroke prevention and it costs much less than stroke treatment.

## 2. Materials and Methods

### 2.1. Study Design, Setting, Population, Sample Size, and Sampling Method

This PAR study was conducted in the two provinces with the leading stroke incidences in southern Thailand out of five in total, i.e., Trang (597.4 per 100,000) and Phatthalung (648.3 per 100,000) [15]. The villagers in three villages of Phatthalung province (Tamode, Klong Yai, and Rawangkuan) and one village of Trang province (Tatomek) were invited to participate in this study after study information was provided. Twenty-seven people per village were required for sufficient statistical power according to the sample size calculation website (GPOWER) to compare the differences between two dependent means (matched pair, effect size 0.5; power 0.8). Therefore, we planned to enroll at least 30 participants aged 45 years and above from each village.

Initially, the study participants were evaluated for their level of stroke risk using the Stroke Risk Assessment tool designed and validated by the Department of Disease Control, Ministry of Public Health, Thailand (DDC-TMOPH) [33]. The stroke risk levels were classified as low, medium, or high. Later, we further invited the study participants whose stroke risks were classified as medium or high to participate in the next steps of the study, in which the co-designed behaviors were applied. We excluded villagers who had a previous history of stroke and those who were unable to perform the co-designed behaviors and activities because of physical disabilities. Participation in both steps of the stroke risk assessment and the co-designed behaviors study was voluntary. Signed written consents to participate in the individual steps of the study were obtained from all invitation-accepting study participants.

Concerning the study locations, Phatthalung province is located on the eastern coast of the Malay peninsula, while Trang is on the western coast. However, there are no significant differences between the two provinces in terms of the local people’s lifestyles, cultures, or livelihoods.

### 2.2. Study Tools and Definitions

a. The Stroke Risk Assessment tool is a simple tool for classifying stroke risk levels. The tool was developed and validated by the DDC-TMOPH. The parameters considered for scoring in the tool were (1) age > 35 years, (2) history of stroke among close relatives, (3) BMI > 25 and WC > 90 cm in males or >80 cm in females, (4) having DM, (5) having HT, (6) having DLP, and (7) current active or passive smokers. In this study, participants who met 0–1 item, 2–4 items, or 5–7 items were classified as having a low, medium, or high stroke risk level, respectively [33].

b. Psychological stress is defined as a condition in which psychological pressures cause an individual to have an unhappy life. In this study, psychological stress was assessed by the Suanprung Stress Test-20 (SST-20), a psychological assessment tool developed and validated by a team of psychiatrists in Thailand (Cronbach’s alpha reliability coefficient = 0.85). The test consists of 20 questions that evaluated the level of psychological stress experienced by an individual. The severity of psychological stress in each question was categorized into five levels according to the individual’s perception of psychological pressure levels. The scores ranged from 1 (no stress) to 5 (highest stress). For our study, the summation of the scores from all the questions was classified into 0–23 (low stress), 24–45 (moderate stress), 46–61 (high stress), and ≥ 62 (severe stress) [34].

c. Physical measurement tools including a standard weighting machine, height meter, and measuring tape, which were calibrated for anthropometric accuracy by the DDC-TMOPH, were used in the physical health assessments. Blood chemistry tests, i.e., fasting blood sugar and blood lipids, were assessed in the certified laboratory units of the local hospitals in the study areas.

### 2.3. Research Study Process

The researchers initially provided information concerning stroke risk factors, clinical presentation of strokes, stroke outcomes, stroke burden, and how to prevent or minimize the risks of experiencing a stroke to the villagers. Subsequently, the details of the study process were provided to the villagers for their decision to participate in the study voluntarily. After the ethical approval from the institutional review board and signed consent forms were obtained, the participant villagers initially underwent a stroke risk level assessment using the Stroke Risk Assessment tool. Those who were classified as having a medium or high stroke risk were further invited to participate in the next steps of the study, in which co-designed health behaviors were involved.

The PAR approach in this study started with the organization of a series of group meetings. The participants of these group meetings included the villagers who accepted the further invitations as the ‘study participants’, local HCPs, community leaders, religious leaders, cooks who frequently cooked meals at social meetings in the studied communities, local grocery shop owners, and the research team. At least five to ten people from each group were invited to attend the meetings. The group meetings were conducted by the researchers (C.S. and J.T.) to lead discussions regarding risk behaviors or attributable factors of elevated stroke incidence in the individual villages. The study participants were encouraged to freely share their comments and suggestions of necessary practicable healthy behaviors as self-reliant ways to control the stroke risk factors. During this step, we expected that the knowledge about stroke risk factors provided to the study participants at the beginning and their understanding of the actually available health resources in their communities could drive them to generate ideas concerning reasonable, mindful, and moderate utilization of the health resources available as well as to realize the benefits of practicing self-care for controlling stroke risk factors (Figure 1).

Finally, the consensus on the co-designed health behaviors for controlling stroke risk factors included dietary control, regular exercise, psychological stress reduction, and reducing or stopping smoking and alcohol consumption. The details of the individual co-designed modified behaviors based on the consensus of the group meetings are as follows:-Dietary control: The group meeting consensus suggested in-house cooking for low-calorie, low-salt, and low-sugar meals according to standard dietary advice for reducing the risk of cardiovascular diseases endorsed by public health agencies. High dietary fiber food could be accessible from the vegetables and grains grown in the communities. The study participants were advised to exchange or share their harvested crops with the other community members instead of buying them from the markets for reducing household expenses. It was also advised that cooking for social meetings in the communities should follow the standard dietary guidelines.-Regular physical exercise: A 20- to 30-min evening walk ≥3 times/week was the regular exercise proposed by the study participants. As most of them woke up very early after midnight (around 2–3 AM) to work at their rubber plantations, they did not have enough time to perform morning exercise. In addition, because the participants were mainly older adults, walking was considered suitable for their physical status.-Psychological stress reduction: This involved joining Buddhist or other religious activities such as praying, offering food to monks, or making merit in other ways according to the individual’s religious beliefs and teachings. These were believed to calm one’s mind and relieve psychological stress.-Reduction in or cessation of smoking and alcohol consumption: Reducing or completely stopping smoking and alcohol consumption was encouraged.

### 2.4. Data Collection and Analysis

a. Physical measurements (BW, height, BMI, WC, and blood pressure (BP)) and blood tests (FBS, total cholesterol (TC), triglycerides (TG), HDL cholesterol (HDL-C), and LDL cholesterol (LDL-C)) were performed for the baseline data. The SST-20 was also applied to evaluate the initial level of psychological stress of the study participants.

b. When the co-designed health behaviors for controlling stroke risk were initiated, the study participants were encouraged to record the frequencies of performing the co-designed behaviors in a week and report them to the local HCPs, who served as research assistants of this study. The reported frequencies of performing each individual behavior were averaged into times/week and transferred to the research team during monthly field visits for statistical analysis. The levels of adherence to the co-designed health behaviors were classified according to the frequency of performing the individual behavior per week, i.e., >5/week = 3 points, 3–4/week = 2 points, 1–2/week = 1 point, and 0 = 0 points. The summation of points obtained from each behavior was averaged and further categorized as favorable adherence if the averaged points amounted to 2.01–3.00, moderate if 1.01–2.00, or unfavorable if 0–1.00.

c. We remeasured the BW, BMI, WC, BP, and blood profiles of the study participants at the end of the program for comparison with the baseline results.

d. Descriptive statistics, i.e., frequencies, percentages, means (SD), medians (IQR), and paired t-tests, were used for statistical analysis of significant differences (*p* < 0.05) using the R program version 4.3.3 [35].

## 3. Results

A total of 126 participants were enrolled from three villages in Phatthalung province (Tamode, Klong Yai, and Rawangkuan) and one village (Tatomek) in Trang province. All study participants were ≥60 years old, and 75.4% of them were female. They had been diagnosed with and received medicines for DM in 5 cases (3.9%), DLP in 39 cases (30.9%), and HT in 51 cases (40.4%). However, they were unaware of the benefits of health behavior modifications in addition to pharmacological treatments for stroke prevention.

The general characteristics of the participants are shown in Table 1. The risk level for stroke among the study participants assessed before the initiation of the health behavior program was classified as moderate in 58.2% and high in 19.8% of the participants (Table 2).

The baseline physical measurements and blood test results revealed that 50–80% of the study participants had an elevated WC (>80 cm for females; >90 cm for males), BMI (≥23), BP (systolic BP (SBP) ≥ 140 and/or diastolic BP (DBP) ≥ 90 mmHg), TC (≥200 mg/dL), TG (≥150 mg/dL), and LDL-C (≥100 mg/dL) (Table 3).

The frequencies of adherence to following the co-designed health behaviors among the study participants in individual villages and in all four villages during this study were significantly higher than before the study program initiation. However, when individual health behaviors were evaluated, only dietary control showed significant adherence among the four study villages. Regular physical exercise in all four study villages and psychological stress reduction practices in Klong Yai showed no significant adherence.

At the end of the program, BP and blood lipid test results in all villages showed significant reductions. Moreover, LDL-C significantly decreased and HDL-C significantly increased in all villages. These were beneficial outcomes expected from favorable adherence to the program. Only WC did not show a significant decrease among the physical measurements. However, when individual villages were evaluated, DBP in one village (Tatomek), FBS in three villages (Tamode, Rawangkuan, and Tatomek), and TC in one village (Klong Yai) showed no significant improvements. The changes in BP and results of blood tests after the program indicated the benefit of the co-designed behaviors on controlling the stroke risk factors. The results concerning smoking and alcohol consumption were too limited to be analyzed.

## 4. Discussion

Most of the participants in this study were female (75.4%), which might be attributed to higher levels of perception of stroke risk among females. A community healthcare study in Poland had a similar finding that women had a higher rate of participation and adherence to cardiovascular health advice concerning cardiovascular risks control than men, especially advice related to risk behaviors and dietary control [36]. At the baseline evaluation of our study, nearly one-fifth of the study participants had a high stroke risk assessed by the Stroke Risk Assessment tool. Additionally, the baseline assessments of WC, BMI, BP, and the results of the blood tests, except for FBS and HDL-C, were elevated in at least half of the participants. The elevation of these health parameters is associated with a high stroke risk. Although a portion of the participants were taking medicines to control these risk factors, most of them were unaware of the benefits of combining behavioral modifications with pharmacological treatment for controlling stroke risk factors. We informed the villagers of their levels of stroke risk from the assessments initially performed, provided knowledge regarding how to reduce their stroke risk, and invited those who had moderate or high stroke risk to join the co-designed behavioral program. After these actions, the study participants could better understand their personal stroke risk and realize the stroke severity or burden they might experience if they had a stroke. Hence, they were motivated to actively participate in and strongly adhere to the co-designed health behaviors of the program. Finally, the improvement in the follow-up results of the physical measurements and blood tests at the end of the program reflected the benefit of adherence to the co-designed behavioral program.

The rationale of applying the PAR method in this study was to facilitate the participation of various local healthcare stakeholders in controlling stroke risk factors and reducing the stroke incidence in the communities. Many studies have found that the PAR method can be a beneficial healthcare strategy in promoting active participation from community members, with cooperation and sustained adherence to a program [31,32,37,38]. Applying the PAR method for the promotion of self-care practices in stroke risk reduction required the provision of sufficient knowledge first, followed by forming positive attitudes and, finally, active practice. Providing knowledge regarding stroke risk and prevention was crucial for the formation of reasonable thoughts (reasonableness) which would facilitate the voluntariness to practice self-care to prevent strokes. Self-care practices could reduce the cost of healthcare management, considered as moderation or economical use of healthcare resources by promoting self-protection behaviors against strokes. The loop of actions mentioned followed the three major elements of the late King’s SEP. Therefore, the SEP concepts and PAR method can be integrated in the context of healthcare management as they share the initial steps of knowledge provision and forming positive attitudes (or reasonable thoughts), followed by self-care [20,32]. Additionally, based on Buddhist or other religious teachings of morality in terms of kindness and compassion, those who achieve the health goals can support and facilitate non-achievers in the community to achieve the same goals also (Figure 2) [24,25,27]. Based on these reasons, we believed that the positive consequences of this integrated approach reinforced by moral thoughts and practices according to religious teachings could be widely distributed to cover the other community members. 

The co-designed health behaviors from the consensus of the group meetings were based on the knowledge provided and practicable among the community members. We considered that this approach would follow the health belief model (HBM) concepts as well [39], in which stroke vulnerability or risk, stroke severity, and the benefits of adopting the co-designed behaviors to prevent a stroke were perceived by the participants, while cues to practice by limiting the existing barriers were proposed by following the consensus in the group meetings. It was noteworthy that while an integrated SEP and PAR approach was carried out, the HBM concepts simultaneously had an impact on the villagers’ behaviors. 

The co-designed health behaviors in this study were mostly suitable for the participants’ levels of physical fitness and lifestyles. This explained why significant adherence to the co-designed behaviors, except for regular physical exercise, was found. It was also suggested in an earlier study that the suitability of the health behaviors or activities of a program was a significant determinant of program adherence and sustainability [40]. Regarding the regular physical exercise that showed no significant adherence in our study, it was possible that there was a misunderstanding that daily agricultural work activities in rice fields or rubber plantations were comparable to physical exercise. This misunderstanding needs additional explanation for correction to the community members in future studies.

In contrast to receiving healthcare services provided by healthcare agencies as usual, we believed that active participatory action motivated by the PAR method combined with SEP concepts would significantly impact program adherence and sustainability, and subsequent favorable outcomes as well. This approach could initiate not only a sense of partnership but also perceivable benefits from practicing self-care among the community members. Earlier studies have confirmed the benefits of encouraging the engagement of community dwellers in developing and operating community-based health promotion programs because of sustainable adherence and the success of such programs [41,42].

When the program ended, the physical measurements and blood tests had significantly improved, except for WC, which could be due to the shorter follow-up time of this study. Therefore, an intense and multidisciplinary behavioral approach to reduce WC, which is a significant indicator of metabolic syndrome and a risk factor for cardiovascular diseases, should be emphasized in future studies with a longer follow-up time.

### Strengths and Limitations

In this study, we demonstrated the significant benefits of a community dweller-centered, lifestyle-compatible, and active participation-driven health program on the initiation and facilitation of self-care practices. We suggested that the SEP concepts combined with the PAR method were applicable for designing a community-based health program under limited health resource circumstances.

As the study was conducted in community settings where healthcare resources were limited and sophisticated facilities were unavailable too, the blood tests performed in this study did not include vascular inflammatory biomarkers specific to atherosclerosis. Moreover, it was our intention to encourage the local HCPs and villagers to use locally available and low-cost but scientifically supported test methods in evaluating and monitoring stroke risk factors, particularly under the context of self-reliant and community-based stroke prevention. Additionally, due to the small sample size, limited study areas, short follow-up time, and limited data on smoking and alcohol consumption, the generalizability of this study will be limited.

## 5. Conclusions

Although SEP was originally an economic philosophy designed to counteract economic constraints and socio-economic disparities in Thai society, it can be integrated with the PAR approach in healthcare management. Initiation of self-care practices to lower stroke risks with lower health budget requirements is the expected main outcome of the SEP and PAR integrated approach in healthcare. Future studies with larger sample sizes, different study locations, diverse socio-cultural contexts, and various diseases preventable by behavioral modifications are required to confirm the benefits and generalizability of the integrated approach proposed by this study.

## Figures and Tables

**Figure 1 healthcare-12-01367-f001:**
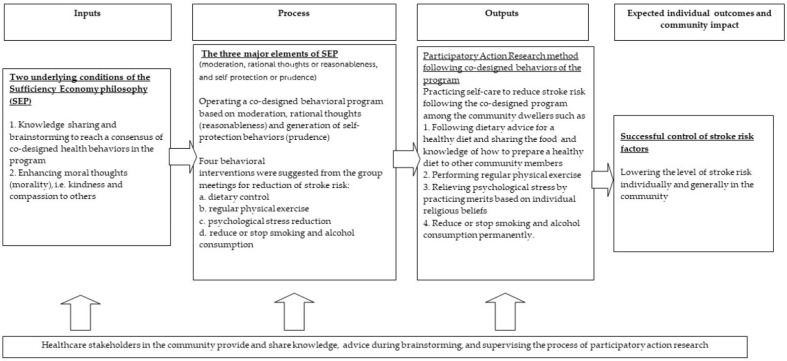
Input, process, output, and expected outcomes of a combined SEP and PAR method in a co-designed health behaviors program for stroke risk factors control.

**Figure 2 healthcare-12-01367-f002:**
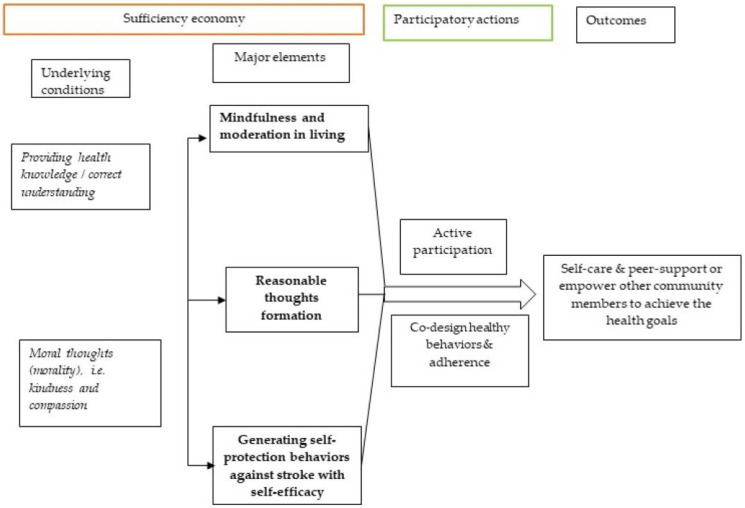
Illustration of the combined application of the sufficiency economy philosophy (SEP) and the participatory action research (PAR) method to encourage active participation, practicing self-care, and supporting community healthcare.

**Table 1 healthcare-12-01367-t001:** General characteristics of the study participants.

Variable	Villages of Phatthalung Province, *n* (%)			Village of Trang Province, *n* (%)	Total, *n* (%)
	Tamode (*n* = 33)	Klong Yai (*n* = 32)	Rawangkuan (*n* = 32)	Tatomek (*n* = 29)	
Gender					
Male	6 (18.2)	6 (18.8)	8 (25.0)	11 (37.9)	31 (24.6)
Female	27 (81.8)	26 (81.2)	24 (75.0)	18 (62.1)	95 (75.4)
Age, years					
60–70	29 (87.9)	28 (87.5)	25 (78.1)	19 (65.5)	101 (80.2)
71–80	4 (12.1)	3 (9.4)	5 (15.6)	10 (34.5)	22 (17.5)
80+ Level of education	0 (0)	1 (3.1)	2 (6.3)	0 (0)	3 (2.4)
None	4 (12.1)	5 (15.6)	7 (21.9)	10 (34.5)	26 (20.6)
Primary school	24 (72.7)	24 (75.0)	20 (62.5)	17 (58.6)	85 (67.5)
Secondary school or higher	5 (15.2)	3 (9.4)	5 (15.6)	2 (6.9)	15 (11.9)
Occupation					
Housewife	9 (27.3)	11 (34.4)	13 (40.6)	1 (3.4)	34 (27.0)
Farmer/gardener	23 (69.7)	19 (59.4)	18 (56.3)	9 (31.1)	69 (54.8)
Fishery	0 (0.0)	0 (0.0)	0 (0.0)	19 (65.5)	19 (15.1)
Others	1 (3.0)	2 (6.2)	1 (3.1)	0 (0.0)	4 (3.1)
Religion					
Buddhism	31 (93.9)	29 (90.6)	32 (100)	23 (79.3)	115 (91.3)
Islam	2 (6.1)	3 (9.4)	0	6 (20.7)	11 (8.7)
Income (THB/month)					
<5000	12 (36.4)	14 (43.4)	17 (53.0)	19 (90.6)	62 (49.2)
5000–10,000	17 (51.5)	17 (53.5)	11 (34.5)	8 (9.4)	53 (42.1)
>10,000	4 (12.1)	1 (3.1)	4 (12.5)	2 (0.0)	11 (8.7)
Smoker					
Yes	4 (12.1)	4 (12.5)	4 (12.5)	9 (31.0)	21 (16.7)
No	29 (87.9)	28 (87.5)	28 (87.5)	20 (69.0)	105 (83.3)
Alcohol consumption					
Yes	10 (30.3)	6 (18.7)	5 (15.6)	8 (28.6)	29 (23.8)
No	23 (69.7)	26 (81.3)	27 (84.4)	21 (71.4)	97 (76.2)
Total	33 (100.0)	32 (100.0)	32 (100.0)	29 (100.0)	126 (100.0)

**Table 2 healthcare-12-01367-t002:** Baseline levels of stroke risk among the study participants before the participatory action program initiation.

Level of Stroke Risk	Villages of Phatthalung Province, *n* (%)			Village of Trang Province, *n* (%)	Total, *n* (%)
	Tamode (*n* = 33)	Klong Yai (*n* = 32)	Rawangkuan (*n* = 32)	Tatomek (*n* = 29)	
Low (0–1 items)	5 (20.0)	3 (13.0)	7 (31.9)	5 (23.8)	20 (22.0)
Moderate (2–4 items)	14 (56.0)	12 (52.2)	13 (59.1)	14 (66.7)	53 (58.2)
High (5–7 items)	6 (24.0)	8 (34.8)	2 (9.0)	2 (9.5)	18 (19.8)
Total	25 (100.0)	23 (100.0)	22 (100.0)	21 (100.0)	91 (100.0)

**Table 3 healthcare-12-01367-t003:** Baseline physical measurements and blood test results of the study participants.

Variable	Villages of Phatthalung Province, *n* (%)			Village of Trang Province, *n* (%)	Total, *n* (%)
	Tamode (*n* = 33)	Klong Yai (*n* = 32)	Rawangkuan (*n* = 32)	Tatomek (*n* = 29)	
WC					
Normal	7 (21.2)	5 (15.6)	8 (25.0)	7 (24.1)	27 (21.4)
Abnormal (>80 F, >90 M)	26 (78.8)	27 (84.4)	24 (75.0)	22 (75.9)	109 (78.6)
BMI					
Normal (18.5–22.9)	3 (9.1)	4 (12.5)	6 (18.8)	5 (17.2)	18 (14.3)
Overweight (23–24.9)	3 (9.1)	3 (7.4)	2 (6.3)	3 (10.3)	11 (8.7)
Obese level 1 (25–29.9)	15 (45.1)	14 (43.8)	11 (34.4)	9 (31.1)	49 (38.9)
Obese level 2 (>30)	12 (36.7)	11 (34.4)	13 (40.6)	12 (41.4)	48 (38.1)
BP (SBP, DBP mm.Hg.)					
Normal (120–129, and/or 80–84)	7 (21.2)	5 (15.6)	4 (12.5)	6 (20.8)	22 (17.5)
High normal (130–139, and/or 85–89)	12 (36.4)	16 (50.0)	10 (31.3)	15 (51.7)	53 (42.1)
Grade I HT (mild) (140–159, and/or 90–99)	8 (24.1)	3 (9.4)	11 (34.4)	5 (17.2)	27 (21.4)
Grade 2 HT (moderate) (160–179, and/or 100–109)	2 (6.1)	5 (15.6)	7 (21.8)	3 (10.3)	17 (13.5)
Grade 3 HT (severe) (≥180, and/or ≥110)	2 (6.1)	1 (3.1)	0 (0.0)	0 (0.0)	3 (2.4)
Isolate systolic HT (≥140, and <90)	2 (6.1)	2 (6.3)	0 (0.0)	0 (0.0)	4 (3.2)
FBS (mg%)≤110 mg%	23 (69.7)	28 (87.5)	28 (87.5)	24 (82.8)	110 (87.3)
>110 mg%	10 (30.3)	4 (12.5)	4 (12.5)	5 (17.2)	16 (12.7)
TC (mg%)					
<200	10 (30.3)	17 (53.1)	12 (37.5)	8 (27.6)	47 (37.3)
200–239	14 (42.4)	7 (21.9)	9 (28.1)	10 (34.5)	40 (31.7)
>240	9 (27.3)	8 (25.0)	11 (34.4)	11 (37.9)	39 (31.0)
LDL-C (mg%)					
<100	6 (18.2)	7 (21.9)	9 (28.1)	5 (17.2)	27 (21.4)
100–129	9 (27.3)	11 (34.4)	8 (25.0)	10 (34.5)	38 (30.2)
130–159	9 (27.3)	6 (18.8)	5 (15.6)	7 (24.1)	27 (21.4)
160–189	7 (21.2)	5 (15.6)	6 (18.8)	3 (10.3)	21 (16.7)
>190	2 (6.1)	3 (9.4)	4 (12.5)	4 (13.8)	13 (10.3)
TG (mg%)					
<150	19 (57.6)	19 (59.4)	17 (53.1)	10 (34.5)	65 (51.6)
150–199	6 (18.2)	5 (15.6)	9 (28.1)	7 (24.1)	27 (21.4)
200–499	8 (24.2)	8 (25.0)	6 (18.8)	12 (41.4)	34 (27.0)
>499	0 (0.0)	0 (0.0)	0 (0.0)	0 (0.0)	0 (0.0)
HDL-C (mg%)					
>60	25 (75.8)	15 (46.9)	25 (78.1)	23 (79.3)	88 (69.8)
40–59	8 (24.2)	17 (53.1)	7 (21.9)	6 (20.7)	38 (30.2)
<40	0 (0.0)	0 (0.0)	0 (0.0)	0 (0.0)	0 (0.0)

Abbreviations: BP, blood pressure; SBP, systolic blood pressure; DBP, diastolic blood pressure; FBS, fasting blood sugar; TC, total cholesterol; HDL-C, high-density lipoprotein cholesterol; TG, triglycerides; LDL-C, low-density lipoprotein cholesterol; BW, body weight; WC, waist circumference; HT, hypertension.

## Data Availability

All the study data, analysis methods, and results generated from this study are shown in this published article. No data or any parts of them were deposited in any preprint servers or online websites.

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
