# Peer review of "Integrating the Sufficiency Economy Royal Philosophy and Participatory Action Research Approach to Promote Self-Care for Stroke Prevention in Selected Communities of Southern Thailand"

_healthcare, 2024, doi:10.3390/healthcare12141367_

Round 1

Reviewer 1 Report

Comments and Suggestions for Authors

This study shows how combining the sufficiency economy philosophy (SEP) with participatory action research (PAR) can help people in southern Thailand villages take better care of themselves to reduce the risk of stroke. The study lasted eight months and involved villagers who were at moderate to high risk of having a stroke. The researchers found that when people followed the health advice they helped create, their blood pressure and other stroke risks went down significantly. This proves the study's main idea. This approach could be a useful way to improve the lives of people with stroke symptoms by focusing on preventing strokes rather than just treating them. Below, please, find a few questions and suggestions for the authors:

     In the manuscript's discussion section, authors are encouraged to elucidate the rationale behind examining waist circumference (WC), body weight (BW), body mass index (BMI), blood pressure (BP), fasting blood sugar level, blood lipid profile, and smoking and alcohol consumption in relation to their association with stroke occurrence.

    The authors did not provide the previous medication history of selected stroke patients. Were the participants in the study already on stroke-preventive medication or unaware of the possibility of stroke? This detail should be mentioned in the respective section.

    The authors performed a blood test, yet it is unclear whether they identified any inflammatory markers, such as interleukins, which are strongly associated with stroke conditions. Incorporating these results would enhance the manuscript.

Author Response

Please see the file attached.

Reviewer 2 Report

Comments and Suggestions for Authors

Dear Authors,

The study represents a unique research approach that is intended to improve patient health. This ultimately helps you plan and understand the impact of self-care

on the incidence of stroke. The structure and content of the document are well prepared, with particular emphasis on the description of methods and ensuring the repeatability of methodological conditions, as well as obtaining the knowledge necessary to justify the findings. Despite the high level presented, the document still requires some minor modifications related to clarifying aspects and refining the format - minor aspects that certainly do not affect the quality of the document.

If the authors provide scientific evidence in the content, it is worth adding references, such as L112

It seems that the discussion contains too much content regarding the summary, e.g. Line 422. It is worth introducing comparisons from other studies into the discussion.

Author Response

Please the file attached

Reviewer 3 Report

Comments and Suggestions for Authors

1.Revise the abstract to ensure it provides a concise summary of the study, including objectives, methods, results, and conclusions. Consider reducing redundancy and ensuring all key points are clearly addressed.

2.Clarify the flow of information, especially when introducing the sufficiency economy philosophy (SEP) and its relevance to healthcare. Ensure the rationale for combining SEP with participatory action research (PAR) is explicitly stated.

3.Provide more detail on the PAR approach. Clarify the steps taken during the participatory action phase and how community input was incorporated into the program design. Provide more detail on the PAR approach. Clarify the steps taken during the participatory action phase and how community input was incorporated into the program design.

4.Consider summarizing key points in the text and referring to the tables for detailed data. Highlight significant findings clearly in the text, avoiding excessive repetition of numbers already presented in tables.

5.Discuss the implications of the findings in more detail, especially how SEP and PAR contributed to the observed outcomes. Suggest future research directions based on the study’s findings.

6.Use consistent terminology throughout the manuscript. For instance, refer to "SEP" and "PAR" consistently once introduced.

Comments on the Quality of English Language

In general, it is good. A few minor editing will make it better.

Author Response

Please see the file attached

Reviewer 4 Report

Comments and Suggestions for Authors

I find your research interesting and relevant. However, I think it is necessary to expand on the literature on which you rely in order to justify the topic in the introduction. 

You should also expand on the concept of stroke, because research is needed. You can justify this with an interesting article such as:

-          A Cross-Sectional Study: Determining Factors of Functional Independence and Quality of Life of Patients One Month after Having Suffered a Stroke. González-Santos, J (Gonzalez-Santos, Josefa) [1] ; Rodríguez-Fernández, P (Rodriguez-Fernandez, Paula) [1] ; Pardo-Hernández, R (Pardo-Hernandez, Rocio) [1] ; González-Bernal, JJ (Gonzalez-Bernal, Jeronimo J.) [1] ; Fernández-Solana, J (Fernandez-Solana, Jessica) [1] ; Santamaria-Peláez, M (Santamaria-Pelaez, Mirian) [1]

DOI10.3390/ijerph 20020995

You should add the inclusion and exclusion criteria used in your study, as well as the ethics committee.

What software did you use to obtain the results? Also in the statistical analysis you should detail the tests you have carried out to obtain your results.

You also need to expand the discussion, also by comparing with other studies carried out to date and contracting your results.

Author Response

Please see the file attached

Round 2

Reviewer 3 Report

Comments and Suggestions for Authors

Agree to accept in the present form

Author Response

Thank you for your comments and suggestions

Reviewer 4 Report

Comments and Suggestions for Authors

Thank you very much for taking into account the comments made. 

Your manuscript has now improved its scientific quality

Author Response

(The authors gave the same response as above.)
